# Molecular Modeling of EUV Photoresist Revealing the Effect of Chain Conformation on Line-Edge Roughness Formation

**DOI:** 10.3390/polym11121923

**Published:** 2019-11-22

**Authors:** Juhae Park, Sung-Gyu Lee, Yannick Vesters, Joren Severi, Myungwoong Kim, Danilo De Simone, Hye-Keun Oh, Su-Mi Hur

**Affiliations:** 1Department of Polymer Engineering, Graduate School, Chonnam National University, Gwangju 61186, Korea; 2School of Polymer Science and Engineering, Chonnam National University, Gwangju 61186, Korea; 3Department of Applied Physics, Hanyang University, Ansan-si 15588, Korea; 4Department of Advanced Patterning, IMEC, 3001 Leuven, Belgium; 5Department of Chemistry, KU Leuven, 3000 Leuven, Belgium; 6Department of Chemistry, Inha University, Incheon 22212, Korea

**Keywords:** EUV photoresist, line-edge roughness, coarse-grained model, molecular simulation

## Abstract

Extreme ultraviolet lithography (EUVL) is a leading-edge technology for pattern miniaturization and the production of advanced electronic devices. One of the current critical challenges for further scaling down the technology is reducing the line-edge roughness (LER) of the final patterns while simultaneously maintaining high resolution and sensitivity. As the target sizes of features and LER become closer to the polymer size, polymer chain conformations and their distribution should be considered to understand the primary sources of LER. Here, we proposed a new approach of EUV photoresist modeling with an explicit description of polymer chains using a coarse-grained model. Our new simulation model demonstrated that interface variation represented by width and fluctuation at the edge of the pattern could be caused by characteristic changes of the resist material during the lithography processes. We determined the effect of polymer chain conformation on LER formation and how it finally contributed to LER formation with various resist material parameters (e.g., Flory–Huggins parameter, molecular weight, protected site ratio, and *T_g_*).

## 1. Introduction

Extreme ultraviolet (EUV) lithography technology [1,2] is the selected lithographic technology for sub-1x nm patterning [3]. However, important challenges need to be tackled to produce EUV for high-volume manufacturing (HVM), including source power availability, keeping the mask defect free, and inspection of the infrastructure for defects [4,5,6,7,8]. Source power availability for EUV HVM has significantly improved in the last few years. As of now, it has achieved reliable source operation with desired availability [8,9]. However, issues with EUV photoresist, especially line-edge roughness (LER) and line-width roughness (LWR) reduction remain crucial challenges when increasing the resolution or the sensitivity.

Chemically amplified resist (CAR) that allows high throughput with an acid-catalyzed reaction to induce a solubility switch in the photoresist is the most widely investigated material for high-resolution EUV photoresist [10,11,12,13,14]. CAR mainly consists of a polymer matrix of copolymers containing protected functional groups usually blended with a photoacid generator (PAG) and a quencher. These polymers and additives undergo characteristic changes by external factors, such as lights and temperature, in an elaborate reaction mechanism that modifies the material during the lithography processes [2,15]. During the exposure step, part of the resist is exposed to photons (13.5 nm wavelength) that are absorbed by atoms, while the electrons are excited and emitted. During elastic and inelastic scattering in the matrix, these electrons generate a cascade of secondary electrons, which interact with PAG molecules [16]. This results in the generation of acids (protons) where the material is exposed to light [17,18]. After exposure, a post-exposure bake (PEB) is applied. The generated acids can diffuse in the polymer matrix and catalyze a deprotection reaction, converting the hydrophobic protected functional groups of the polymer matrix into hydrophilic functional groups that are soluble in the aqueous developer. Hence, polymer chains in the EUV exposed area are soluble in an aqueous developer and dissolved during the development step. In such positive-tone EUV photoresist, desired patterns are finally formed by the remaining polymers after development. Spatial fluctuation along the line at the edge of the pattern after development is called line-edge roughness (LER). LER is the result of various physical phenomena and chemical reactions between the components of the resist material that occur throughout the lithography steps. Thus, LER is affected by several complex factors, for example, diverse process parameters, aerial image quality, PEB, and development conditions.

To tackle the challenge of LER and other patterning issues in the lithography process, various computational modeling studies of EUV photoresist have been conducted. Recent EUV photoresist simulations for LER analysis have mostly focused on stochastic effects, adding random variables based on a continuum model. Kozawa et al. [19,20,21,22] have studied acid diffusion behavior and described it with a Gaussian distribution in a continuum or mesoscale level. Mesoscale models incorporating the effects of the component materials of CAR have been developed based on a lattice model including PAG aggregation and the dissolution process [23,24]. However, previous approaches did not include an explicit description of the polymer itself while focusing on stochastic effects. Rarely has the physical description of polymer chains and their movements been reported. Recently, Kim et al. [25,26,27] studied the mechanism of photochemical reactions in an EUV photoresist matrix and the resulting pattern edge morphology with atomic molecular dynamics simulation, although time and length scales are relatively short and small due to the nature of the simulations. Ideally, EUV photoresist simulations should involve the modeling of relevant components, explicitly polymer chains and additives, with a good understanding of complex physical and chemical reaction occurring under the lithographic conditions, at dimension and time scales relevant for patterning. 

In this paper, we proposed a novel molecular approach that describes polymer chain behavior in EUV photoresist. During the exposure and PEB processes, the characteristic variations in the interface between the exposed and unexposed areas were determined using our model. We verified the effect of interface variations on LER formation after development. Our molecular approach and simulation can accurately assess the impact of motions and conformation changes of EUV resist polymer chains on LER formation.

## 2. Model and Methods 

The major limitations of the atomistic model are time and dimension restrictions of the model system (a few nanoseconds and a few nanometers, respectively). Therefore, it is difficult to investigate the molecular phenomena of the current patterning process with real-time scales and pitch sizes. Since we aimed to investigate the chain conformation and its effects on LER formation, we adapted a coarse-grained bead and spring model for the description of polymer chains explicitly and developed a numerical simulation method for a 16 nm half-pitch patterning process, which is schematically illustrated with a conventional point of view in Figure 1a. Figure 1b shows a schematic of the variations in the polymer chain conformations during the lithography processes. The yellow, red, and blue beads represent the polymer backbone, protected sites, and deprotected sites, respectively. As a relevant example, the minimum simulation box size required for the 16 nm half-pitch process was 32 nm in a direction perpendicular to the line pattern. Considering the film thickness of photoresist (30 nm), the system size was set to 32 × 32 × 30 nm with periodic boundary conditions in the *x*- and *y*-directions, while impenetrable walls were used in the *z*-direction to simulate the film structure.

Polymer chains, the main components of EUV CARs, were modeled with discretized Gaussian chains of *N* coarse-grained beads and spring connecting beads. The chains of *N* = 16, 32, 64, and 128 correspond to an EUV resist with molecular weights around 2000 g/mol, 4000 g/mol, 8000 g/mol, and 16,000 g/mol, respectively. Among the *N* beads in a polymer chain, the beads representing the protected sites were randomly selected according to a given ratio of the functional sites in the polymer chain (*f_p_*). The remainder of the beads were assigned as the polymer backbone. At temperature *T*, the bonded energy is given by the following:(1)$$\frac{H_{b}}{k_{B}T}=\frac{3}{2}\frac{N-1}{R_{e}{}^{2}}\sum_{k=1}^{n}\sum_{i=1}^{N-1}b_{k}{}^{2}\left(i\right),$$
where *k_B_* is Boltzmann’s constant, *b_k_(i)* is the bond vector of the *i*th bead in the *k*th chain, and $R_{e}{}^{2}$ is the mean squared end-to-end distance of an ideal chain of *N* = 128. Simulations for the resist of different molecular weights are conducted by varying the number of coarse-grained beads, *N*, while the reference length unit and the averaged bond length are fixed at $R_{e}$ and $b_{0}$ of *N* = 128, respectively. A spin-coated photoresist film was prepared by randomly distributing chains over the simulation box with a fixed bond length of $b_{0}=R_{e}/\sqrt{N-1}$ where *N* = 128. The first bead of each chain is placed randomly in the simulation box, and the position of the next bead along the chain is randomly selected on a sphere of a radius of $b_{0}$, centered at the previous bead position. Monte Carlo (MC) iteration was performed to relax the chain to satisfy Boltzmann distribution, assuming that the post-applied bake (PAB) after spin coating was enough to rearrange the system into its equilibrium state. One snapshot of the prepared system in its relaxed state is shown in Figure 2a, where the red beads represent protected sites (functional groups) and the yellow beads indicate other monomers of the copolymer backbone.

During the exposure step, the protected sites in the exposed region were converted into deprotected functional groups by a photochemical reaction. Photon shot noise, secondary electron blur, and acid diffusion are often cited as the main reasons for the unpredictable distribution of the deprotected sites in the system. In this study, we excluded these effects and attempted to isolate the effects of polymer chain conformation on LER. Thus, PAG and quenchers were not explicitly modeled in the simulation. Instead, deprotection reactions occurring at the PEB step were modeled with the probability provided from the intensity profile of the aerial image generated using commercial software (Sentaurus Lithography from Synopsys, 2018.06-SP1). When an ideal aerial image was used, assuming a perfectly controlled photochemical reaction, all beads representing the protected sites (red) in half of the simulation domain were changed into deprotected sites (in blue), as shown in Figure 2b.

Temperature is one of the crucial parameters in simulating the PEB process since the mobility of polymer chains strongly depends on the difference between the glass transition temperature (*T_g_*) of the resist material and the designated bake temperature. If the resist is a high *T_g_* material, the polymer chains freeze during bake. In contrast, if the resist has a lower *T_g_* compared to the PEB temperature, its chains have enough mobility to reach thermodynamically favored states, as shown in Figure 2c. Since deprotected sites are generally strongly hydrophilic, whereas protected groups are hydrophobic, a strong repulsion exists between the protected and deprotected sites. This rearranges chain conformations, thus altering the interface widths between the exposed and unexposed areas.

To predict the interface broadness and the amount of fluctuation, we used a particle-based Monte Carlo (MC) simulation based on a theoretically informed coarse-grained (TICG) model. TICG models have been successfully used to investigate the thermodynamic and structural properties of polymeric melt systems, especially in the directed self-assembly of block copolymer systems [28,29]. In TICG modeling, while the polymer chain is represented by a discretized Gaussian chain, the interaction energy between the particles is represented as a function of the density fields [30,31,32,33]. Non-bonded energy is given by the following: (2)$$\frac{H_{nb}}{k_{B}T}=\int_{V}^{}dr\frac{\sqrt{\bar{N}}}{R_{e}{}^{3}}\left\{\chi N\phi_{A}\phi_{B}+\frac{\kappa N}{2}{\left(1-\phi_{A}-\phi_{B}-\phi_{C}\right)}^{2}\right\},$$
where $\phi_{K}=\rho_{K}/\rho_{0}$ represents the normalized local density of type *K* particles, $\rho_{0}$ is the average density, and $\sqrt{\bar{N}}=\rho_{0}R_{e}{}^{3}/N$ is the average number density of chains of *N* = 128 in the unit volume of $R_{e}{}^{3}$. Particles A, B, and C represent the coarse-grained polymer beads containing hydrophobic protected functional groups, the hydrophilic deprotected site, and other monomers of the copolymer backbone, respectively. The non-bonded Hamiltonian, $H_{nb}$, is represented by two physical parameters, χN and κN, where χN is the degree of repulsion between the protected site and the deprotected site, χ is the Flory–Huggins parameter, *N* is the number of segments per chain, and κ is the inverse of the compressibility parameter. Here, we only considered $\chi N\phi_{A}\phi_{B}$, assuming that the backbone and functional groups had non-preferential interaction, regardless of whether the functional groups were hydrophilic or hydrophobic. We considered this assumption valid because the interaction between protected and deprotected groups was dominant over others. The term proportional to κN is the penalty for the total density deviation from its average melt value, where the finite compressibility of the melt was taken into account [31,32]. In our calculations, we used the values κN = 200 and $\sqrt{\bar{N}}$ = 128 for *N* = 128. We performed MC simulations with single-bead displacement moves only and assumed that the width and fluctuation between the exposed and unexposed area at the interface would be correlated with the roughness.

In the development process, chains containing sufficient numbers of deprotected sites were solvated by developers due to the high solubility of deprotected sites in the developing solvents. At the starting point of the development process, chains that were deprotected above a critical fraction (deprotection fraction = 0.8) were removed and replaced with developing solvent molecules, marked in green in Figure 2d. Attractive interaction between the developer and the deprotected site allowed solvent molecules to diffuse into the photoresist matrix and interact with the chains containing hydrophilic deprotected groups, leading to their solvation (Figure 2e). The non-bonded energy was, thus, updated to include the addition of a developer solvent as follows:(3)$$\frac{H_{nb}}{k_{B}T}=\int_{V}^{}dr\frac{\sqrt{\bar{N}}}{R_{e}{}^{3}}\left\{\chi_{AB}N\phi_{A}\phi_{B}+\chi_{AD}N\phi_{A}\phi_{D}+\chi_{BD}N\phi_{B}\phi_{D}+\chi_{CD}N\phi_{C}\phi_{D}+\frac{\kappa N}{2}{\left(1-\phi_{A}-\phi_{B}-\phi_{C}-\phi_{D}\right)}^{2}\right\},$$
where $\phi_{D}$ and $\chi_{KD}$ represent the normalized density of the developer and Flory–Huggins parameter between each bead (K = A, B, and C) and developer (D), respectively. Here, we assumed a strong repulsive interaction between the protected site and the developer ($\chi_{AD}=1.17$), whereas the deprotected site and developer had strong attractive interactions ($\chi_{BD}=-1.17$) and the interaction between the backbone and developer was ignored. To describe the chain rearrangement due to solvation, simulation with single-bead displacement only was conducted. Chains satisfying the conditions for solvation were replaced with developers, as shown in Figure 2d. To determine the chains to be replaced with solvents, both the averaged solvent density and minimum local solvent density over the grids occupied by each chain were tested. A spatially averaged solvent density above 0.5 and a minimum local solvent density above 0.2 were used as critical values for the replacement. The developing solvents enhanced chain mobility during the development process. The mobility strongly depended on solvent concentration, which was spatially and timely non-uniform as the developers penetrated into the polymer resist. The acceptance probability of the suggested MC moves was chosen as a function of the composition of the solvent as follows:(4)$$P_{acc}=\frac{1}{{\left(1+\phi_{0}^{8}\right)}^{4}{\left(1+\phi_{n}^{8}\right)}^{4}}\min\left[1,exp\left[-\frac{\Delta H}{k_{B}T}\right]\right],$$
where *ΔH* is the difference in total energy between the initial and trial configurations and the weight of the polymer is represented with initial and trial polymer compositions, $\phi_{0}$ and $\phi_{n}$ [34]. After most of the chains in the exposed area were removed by the developer, partially deprotected chains were left, swinging loosely at the interface, as shown in Figure 2e. To simulate the drying process, solvents were removed by replacing solvent beads with air beads. The segregation strength between the residual polymers and those beads was increased to χ = 1.56. Figure 2f shows a snapshot image of the chain conformation at the start of drying of the developing solvents. During the relaxation induced by changing the solvent particles to air, the residual polymers started to aggregate and collapsed, as shown in Figure 2g. These residual polymers contributed to the roughness at the interface. Finally, the LER was defined as 3σ deviation in the line pattern edge. Simulation parameters are summarized in Table 1. 

## 3. Results and Discussion

### 3.1. Exposure and Deprotection Reaction 

Changes in the polymer’s properties and its conformations after each patterning process were investigated using the coarse-grained model. Random copolymers constituted the backbone component and the protected functional sites were initially prepared in their equilibrium state. During the subsequent exposure step, other possible sources of LER were excluded by implementing an ideal image profile (i.e., a digital aerial image having infinite contrast) and assuming a perfectly controlled deprotection reaction (Figure 3a). After the exposure process, all protected functional sites located in the exposed domain (left half side of the simulation box shown in Figure 3a) were deprotected. Figure 3b shows a snapshot of the polymer at the interface when *N* = 64 and *f_p_*= 0.5. The yellow, red, and blue beads represent backbone, protected sites, and deprotected sites, respectively. If the chain was located completely within an exposed or unexposed area, the functional sites in the chain were composed of only one type of beads, either protected or deprotected beads. However, at the interfacial region, some chains were partially deprotected, meaning that some beads were changed to deprotected sites, whereas others remained as protected sites. Such chains could have various conformations, as shown in Figure 3b.

After ideal exposure and a well-controlled deprotection reaction, we counted the number of chains in which 40% to 60% of functional sites in the chain are deprotected and reported for different initial polymer chain length (Figure 3c). Although the difference in the concentration of functional sites (protected site and deprotected site) was sharp, a significant number of chains contained both protected and deprotected functional groups. As shown in Figure 3c, the fraction of partially deprotected chains increased as the chain length *N* increased. The partially deprotected chains were located mainly across the interface between the exposed and unexposed areas with a characteristic length of *R_e_*. Although no stochastic effect from exposure or acid diffusion existed, changes in the spatial distribution and conformation of the partially deprotected chains during PEB and the development process induced a non-negligible line-edge roughness. This is discussed further in the following sections.

### 3.2. Chain Conformation Changes during the PEB Process

When the glass transition temperature (*T_g_*) of resist film is lower than the PEB temperature, polymer chains have enough mobility to rearrange into thermodynamically favored states during the PEB process (Figure 2c). Note that the actual *T_g_* of a photoresist film is different compared to a bulk polymer film. The additives in the photoresist film act as a plasticizer and reduce the *T_g_* of the film drastically. Additionally, the thin film geometry generally leads to a decrease of the *T_g_* [35,36], except for a few reported cases in which a strong interaction of resist–substrate led to an increase of *T_g_* [37]. We investigated how chains in the resist with low *T_g_* redistributed during the PEB process due to large repulsive forces between the hydrophilic deprotected and hydrophobic protected sites. We also examined the interfacial width and its fluctuations. Interfacial widths show how broadly functional sites are distributed after the PEB process. Fluctuations describe variability in the position of the center of interfaces between protected and deprotected sites. In contrast, for the resist of high *T_g_*, chain rearrangement is not conducted, and only deprotection reaction is considered during PEB due to the limited motion of polymer chains. The rearrangement of polymer in low *T_g_* resist during the PEB process affects the final roughness at the edge of the pattern which is further discussed in Section 3.3.

It is well known that the interface between the immiscible polymer blends has a finite width, which scales proportionally to the Flory–Huggins parameter between two polymers χ with an exponent of −0.5 ($\mathrm{width}\;\propto 1/\sqrt{6\chi }\;$) [38,39]. EUV CAR can be modeled with a random copolymer composed of backbone materials with and without functional groups. It is expected to have a broader width between the majority of the protected and unprotected domains than a blend of immiscible homopolymers. To understand how the chains were redistributed during the PEB process, we measured the interfacial width and fluctuation of the interface center position at equilibrium state as a function of χ between the different functional groups with fixed *N* = 128 and *f_p_* = 0.5, as shown in Figure 4a. Even with an aerial image with infinite contrast (e.g., diffraction-free), we observed that this interface had a finite width and fluctuated. The segregation strength between the protected and deprotected sites (χ) significantly affected the interfacial width and magnitude of the interface position fluctuation. When a material with higher χ was considered, the interfacial width and amount of fluctuation were smaller than photoresist with a lower χ, although they were still much larger than those of a corresponding homopolymer blend with the same χ.

Further investigations were conducted at fixed χ = 0.78 (χN = 100 for *N* = 128), thus assuming a strong repulsion between hydrophilic deprotected functional sites and hydrophobic protected sites. Figure 4b,c shows variations in the interfacial widths and fluctuations of the interface position as a function of *f_p_* for different *Ns*. All samples, regardless of *N*, presented qualitatively similar interface behaviors as the ratio of the functional sites in the chain (*f_p_*) was increased. The width and magnitude of the position fluctuation of the interface decreased due to enhanced demixing of the two different functional groups at the interface. Note that larger width and fluctuation after PEB did not necessarily result in large final LER, since the chain conformation still changed during the development process as further discussed in Section 3.3. Figure 4d replots the data of *f_p_* = 0.5 from Figure 4b,c to show the dependency of interfacial width and position fluctuation as a function of the chain length of *N*. The system of shorter chains resulted in a broader interface and considerably more fluctuation at the interface. One can naively predict that a photoresist of lower molecular weight polymers is better for reducing the roughness because each single chain occupies a smaller space. Thus, the roughness formed by the empty space after chain removal is also small. However, shorter chains have enhanced mixing entropy and chain mobility. Thus, the interface between the exposed and unexposed domains becomes wider, increasing the final LER. When *N* is higher than a certain value (*N* = 64 in our simulation model), the probability of having partially deprotected chains at the interface increases and the average length of repeated deprotected functional sites along the chain contour, uninterrupted by protected groups, becomes much shorter. This connectivity of protected and deprotected groups enhances the miscibility between them. Hence, there is an optimal chain length that minimizes the uncertainty on the interface position. 

Detailed chain conformations at the interface need to be studied to characterize the source of the roughness from a molecular point of view. When chains are aligned in a direction parallel to the interface, rather than perpendicular, less roughness is expected. We analyzed the chain alignment near the interface (16 ± 1 nm) by measuring the averaged end-to-end distance (*R_e_*) from directions parallel (∥) and perpendicular (⊥) to the interface. Figure 5 shows the average ratio of $R_{⊥}$ to $R_{∥}$, as a function of *N*. Up to *N* = 64, the elongation of perpendicular distribution was significantly decreased. However, further increases in *N* resulted in more spherical conformations, rather than elongated ones. The overall plot matched well with interfacial width and fluctuation (Figure 4d), indicating that chain conformation was correlated to the roughness.

### 3.3. LER Formation after Development

The final roughness at the pattern edges after development was calculated from the molecular-based simulations described in the Model and Methods section. During the development process, developing solvent enhances the polymer chain’s mobility due to plasticizing effects and interacts strongly with hydrophilic deprotected sites along the chains. Thus, polymer chains rearrange to be surrounded by a shell of solvent molecules, and chains that are completely solvated can be easily dissolved. Partially deprotected chains near the interface between the exposed and unexposed domain can be incompletely surrounded by developing solvent molecules and remained undissolved. These partially swollen chains collapsed to the interface during the removal of solvent (drying), generating the final LER. We tracked how chain conformation varied throughout the development process and measured the final LER. Even polymer chains having a high *T_g_* whose conformation remained unchanged during PEB would have an enhanced mobility during the development process, allowing conformational changes. We compared the final LERs after development for EUV resists with high *T_g_* and of low *T_g_* (where polymer chains could rearrange during PEB). 

Figure 6a shows the predicted LER as a function of *f_p_* while *N* was fixed at 64. The red curve represents the LER of a low *T_g_* resist, whereas the blue corresponds to a high *T_g_* resist. There was no significant difference in LER between the high *T_g_* and low *T_g_* systems, except for when *f_p_* = 0.3. When *f_p_* was larger than 0.3 and *N* = 64, the interfacial width for the system of low *T_g_* after PEB was predicted to be less than 3 nm. This value decreased when *f_p_* increased (Figure 4b,c). The final roughness was about 3 nm, which was larger than the magnitude of the interfacial width and fluctuation after PEB. This implies that the development process plays a critical role in LER formation, potentially more than the PEB process. When the system parameters were *f_p_*= 0.3 and low *T_g_*, a broader interfacial width was observed (≈5 nm) after PEB due to the small extent of demixing between the two functional groups (Figure 4b), resulting in a larger final LER after development. In other words, in a high *T_g_* system with a sharp density profile of functional groups (due to the absence of chain rearrangement during the PEB step), a lower LER was observed compared to that of a low *T_g_* system. 

Figure 6b summarizes the predicted LERs as a function of *N* for fixed *f_p_* = 0.5. The results showed that *N* had a significant impact on LER. This indicates that molecular weight was more strongly correlated with LER than *f_p_*. When *N* was large, LER was independent of *T_g_*, which suggests that LER is mainly determined during the development process. The longer the chain length, the larger the space occupied by the residual chains collapsing to the interface after drying. The sizes of these chains can directly impact the magnitude of the LER. However, when *N* is small (*N* < 64), the final LER varies depending on *T_g_*, suggesting that chain rearrangements according to thermodynamic factors during PEB have more impacts on LER than the size of the polymer. When the *T_g_* of the resist is much lower than the PEB temperature, the deprotected chains are broadly distributed over the interface. Some of these can diffuse into the unexposed area due to entropy and enhanced mobility of the short chains during the PEB step. Hence, deprotected chains cannot be removed completely at the interface during the development step, not because of the partially deprotected chains, but due to the non-deterministic distribution of the deprotected chains. This results in increased LER, even though the space occupied by each chain was small. In contrast, high *T_g_* CAR showed lower LER because the chains were frozen during the PEB process. These results also suggest that the use of polymer resin exhibiting high enough *T_g_* at low molecular weight regime could improve the pattern quality. Although we remind the reader that the ideal acid diffusion during the PEB was assumed in our modeling, acid diffusion would be decelerated in a high *T_g_* resist. Figure 6c,c shows 3D images of the residual polymer chains after the development process in a system of low *T_g_* EUV resist with various values for *f_p_* and *N*. Our simulation confirmed that the polymer chain conformations significantly affected the formation of the final roughness, and even though all other contributors (e.g., stochastic terms) were excluded, a certain amount of the final roughness was inevitably induced by the polymer chain conformation.

### 3.4. Effect of the Aerial Image on Polymer Chain Conformation

In the previous sections, we demonstrated that polymer chain conformations significantly correlated to LER formation in EUV CAR. Even when an ideal aerial image profile was implemented, excluding other possible sources of LER such as photon shot noise, acid diffusion, and inhomogeneous resist material distribution, LER was predicted to be larger than 2 or 3 nm, and dependent on χN*, f_p_,* and *N*. In this section, we considered an aerial image profile with a realistic contrast of 16 nm and 1:1 line and space patterning, as shown in Figure 7a (obtained using Sentaurus Lithography). It is equivalent to a strong dipole illumination with a system of NA = 0.33, giving a normalized image log slope (NILS) of 2.8. After exposing the resist to this aerial image, we monitored the proportion of partially deprotected chains (40% to 60% of the hydrophobic functional groups in a chain were converted into hydrophilic groups), as shown in Figure 7b. This aerial image should be compared to the ideal aerial image in Figure 3c. When a diffracted aerial image was imposed, the deprotection reaction occurred over a 3-fold broader region than the ideal aerial image case. It resulted in larger numbers of chains that were partially deprotected.

After exposing the resist to the ideal and diffracted aerial images, the distribution of the protected sites over the resist was compared, as shown in Figure 8a. In the case of the ideal aerial image, the extent of interface mixing of the functional groups between the exposed area and unexposed areas was small, reflecting the infinite contrast of the aerial image. On the other hand, the extent of mixing at the interface was slightly larger when the contrast of the aerial image was about 90%. Obviously, in the latter case, the interfacial width measured after PEB step was broader because of the wider distribution of deprotected sites. The effect of *N* on LER formation for high *T_g_* CAR exposed to a defocused aerial image is shown in Figure 8b. LER formation, as a function of *N*, behaved in a quantitively similar manner in the ideal aerial image (Figure 6b) and realistic aerial image. However, the LER values were significantly higher when the resist was exposed to the non-ideal aerial image; when *N* was higher than 64, bridges were formed and specific LER values were thus not assigned in Figure 8b. The deprotection reaction sites in the interface region were more broadly distributed when the non-ideal aerial image was implemented.

## 4. Conclusions

A theoretically informed coarse-grained model was adapted to study the LER of EUV resist from a molecular aspect. The coarse-grained model allowed for the investigation of molecular phenomena that take place during the 16 nm patterning process. Here, we examined the changes in distribution and conformations of the polymer chains at the interface between the exposed and unexposed areas during the PEB and development processes. Even though we used an ideal aerial image and assumed a perfectly controlled deprotection reaction, excluding other feasible sources of LER, the size of the simulated final pattern roughness was 1.8 nm to 6.3 nm, which is comparable to reported experimental values [40,41]. Our simulation indicates that the polymer chain conformation, which was not considered in previous works, is a non-negligible source of LER and emphasizes the importance of modeling polymer chains explicitly for patterning processes.

The interfacial width and LER of the final patterns were measured by varying physical values of the resists, such as Flory–Huggins parameter (*χ*), degree of polymerization (*N*), and ratio of the protected sites in the chain (*f_p_*) and glass transition temperature (*T_g_*). While LER was not strongly sensitive to the protected site ratio *f_p_*, our simulations showed that the difference in *T_g_* and PEB temperature played an important role in LER. When the resist *T_g_* was higher than PEB temperature, the chain length was a dominant factor for LER, and low molecular weight resist outperformed a high molecular weight system. However, when the PEB temperature was close to the *T_g_* of the resist, chain rearrangement during the PEB process broadened the interface and increased the final LER. Chains with smaller numbers of beads more readily rearranged, hence there exists an optimal chain size for minimizing LER. The simulation results with non-ideal aerial images emphasized that the profile of the deprotection reaction was a crucial factor in LER. For a more elaborate theoretical model of the lithography process, our model should reflect both polymer chain conformation and stochastic terms (e.g., photon shot noise and acid diffusion) in the future. The suggested molecular simulations for EUV photoresist are expected to be more useful in designing a material and processes for EUV resists on the sub-1x nm patterning, where the critical dimension approaches the molecular size.

## Figures and Tables

**Figure 1 polymers-11-01923-f001:**
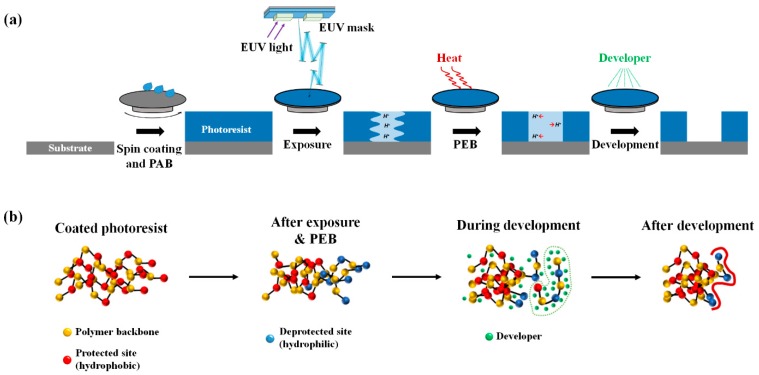
Schematic illustration of the extreme ultraviolet (EUV) lithography process. (**a**) Conventional model of EUV lithography process. (**b**) Our model using coarse-grained molecular simulation. PAB and PEB mean post-applied bake and post-exposure bake, respectively.

**Figure 2 polymers-11-01923-f002:**
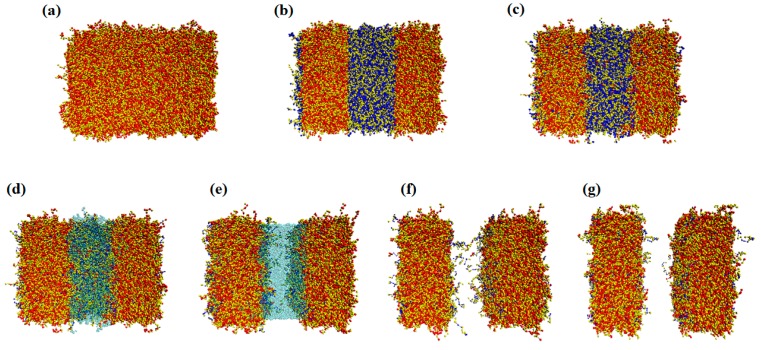
Top view images of the EUV chemically amplified resist (CAR) patterning process. Low *T_g_, N* = 64 and *f_p_* = 0.5, were simulated in this system; (**a**) spin coating, (**b**) exposure, (**c**) post-exposure bake (PEB), (**d**) beginning of the development, (**e**) during development, (**f**) immediately after the development, and (**g**) after drying the remaining developer. Images were generated by wrapping the simulation results in the *x*-direction, corresponding to the system size of 48 × 32 × 30 nm.

**Figure 3 polymers-11-01923-f003:**
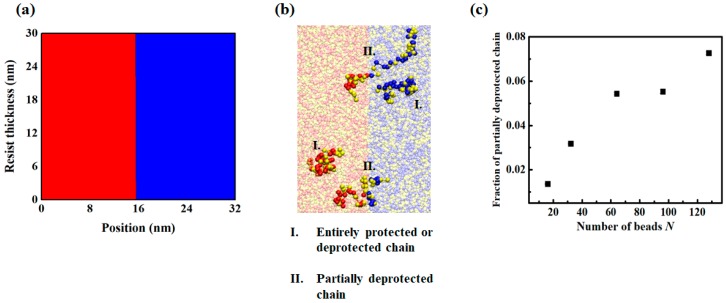
(**a**) Aerial image profile used for an infinite contrast. (**b**) Snapshot of selected chains near the interface; I and II represent the entirely protected or deprotected chains and partially deprotected chains, respectively. (**c**) Fraction of partially deprotected chains in which 40–60% of functional sites are deprotected as a function of the number of beads *N*.

**Figure 4 polymers-11-01923-f004:**
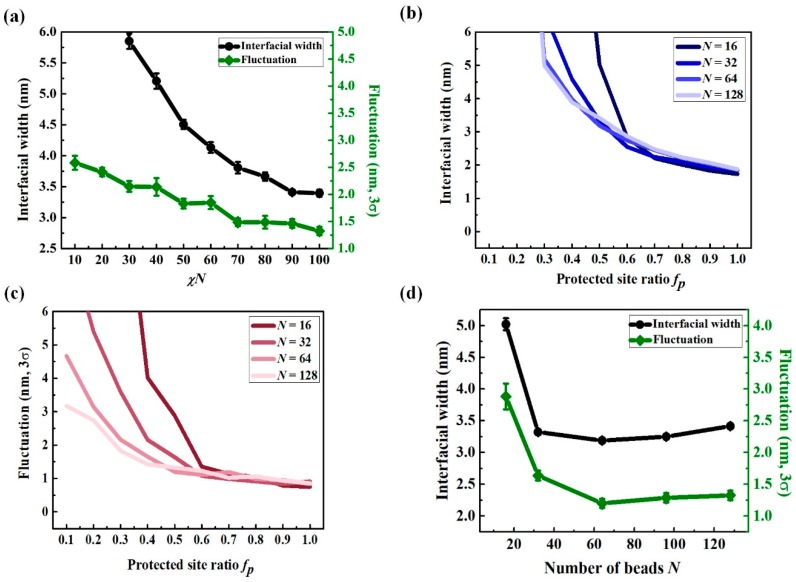
Interfacial width and fluctuation of the interface positions after the PEB process; (**a**) interfacial width and fluctuation as a function of the repulsion degree χN at fixed *N* = 128 and *f_p_* = 0.5, (**b**) interfacial width as a function of the protected site ratio of four different *N*s at fixed χ = 0.78, (**c**) fluctuation as a function of protected site ratio *f_p_* of four different *Ns* at fixed χ = 0.78, and (**d**) interfacial width and fluctuation as a function of *N* at fixed *f_p_* = 0.5 and χ = 0.78.

**Figure 5 polymers-11-01923-f005:**
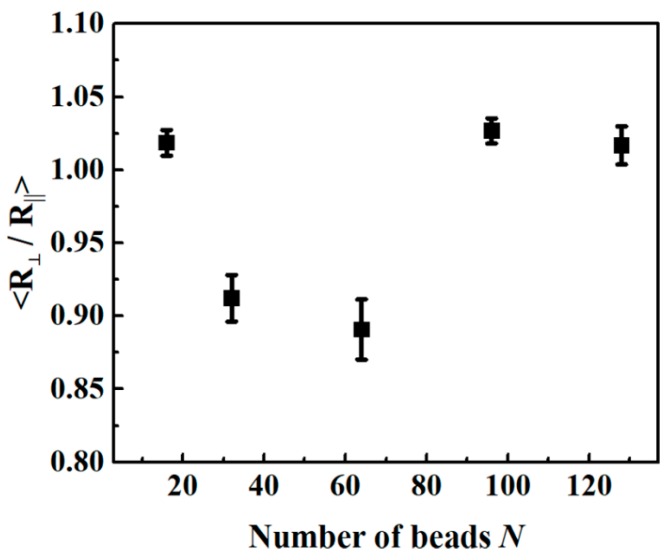
Average ratio of the R_⊥_ to the $R_{∥}$ in the interface region after PEB as a function of the number of beads *N*.

**Figure 6 polymers-11-01923-f006:**
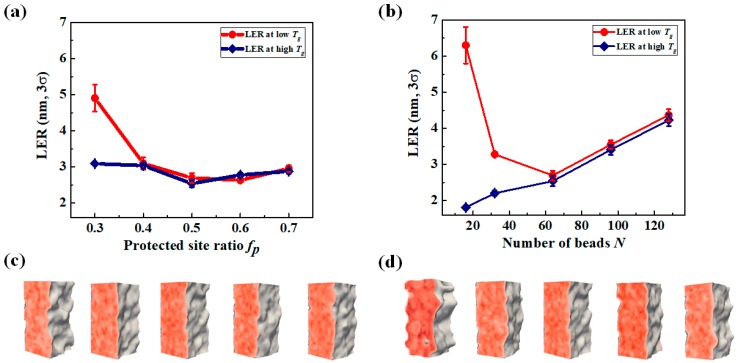
Line-edge roughness (LER) values after the entire CAR patterning process in low *T_g_* and high *T_g_*; (**a**) LER as a function of the protected site ratio *f_p_* at fixed *N* = 64, (**b**) LER as a function of the number of beads *N* at fixed *f_p_* = 0.5, (**c**) 3D image of residual polymer from (**a**) for low *T_g_* resist, and (**d**) 3D image of residual polymer from (**b**) for low *T_g_* resist. In (**c**) and (**d**), red color map and gray contour curve are used to present the variation in the local density of residual resists on cross-cuts and the shapes of line edges, respectively.

**Figure 7 polymers-11-01923-f007:**
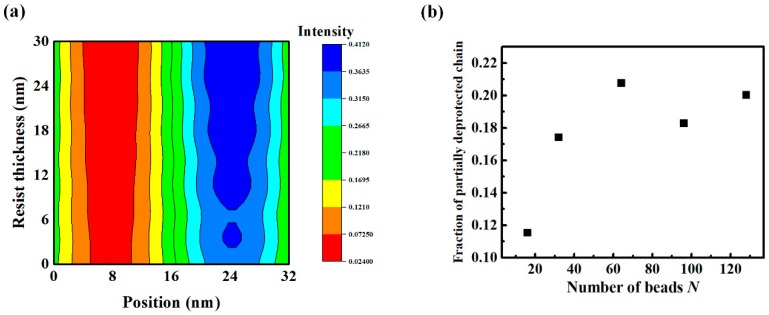
(**a**) Image profile implemented for 16 nm dense line and space (LS), (**b**) fraction of chains which are partially deprotected (40–60% sites deprotected per chain) as a function of the number of beads *N*.

**Figure 8 polymers-11-01923-f008:**
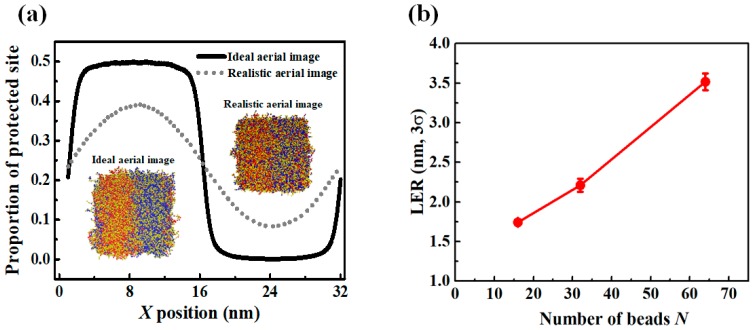
(**a**) Density profile of protected site. The inset shows a snapshot of simulations after exposure step. Left side and right side of inset represent ideal aerial image case and realistic aerial image case, respectively. (**b**) LER values of CAR for high *T_g_* as a function of *N* where 90% contrast image exists with fixed *f_p_* = 0.5.

**Table 1 polymers-11-01923-t001:** Summarized simulation parameters for the system of *N* = 128. A, B, C, D, and E represent protected site, deprotected site, backbone, developing solvent, and air, respectively.

Parameters	Value
$\chi_{AB}N$	100
$\chi_{AD}N$	150
$\chi_{BD}N$	−150
$\chi_{AE}N=\chi_{BE}N=\chi_{CE}N$	200
κN	200
$\sqrt{\bar{N}}$	128

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
