# Peer review of "Molecular Modeling of EUV Photoresist Revealing the Effect of Chain Conformation on Line-Edge Roughness Formation"

_polymers, 2019, doi:10.3390/polym11121923_

Round 1

Reviewer 1 Report

My feeling is definitely good: the manuscript is well organized, clear and well supported. Only few comments in the list below:

Introduction

-line 34 ‘for sub-1x nm patterning’: what is exactly the predicted resolution?

Methods

-Figure 1a and figure 1b: please, improve the quality of the images

Results and Discussion

-Figure 4: please, improve the quality of the images

-line 251              The author claim: “One can naively predict that a photoresist of lower molecular weight polymers is better for reducing the roughness because each single chain occupies a smaller space. Thus, the roughness formed by the empty space after chain removal is also small. However, shorter chains have enhanced mixing entropy and chain mobility. Thus, the interface between the exposed and unexposed domains became wider, increasing the final LER. When N is higher than a certain value (N = 64 in our simulation model), the probability of having partially deprotected chains at the interface increases and the average length of repeated deprotected functional sites along the chain contour, uninterrupted by protected groups, becomes much shorter. This connectivity of protected and deprotected groups enhances the miscibility between them. Hence, there is an optimal chain length that minimizes the uncertainty on the interface position.”

Are there authors who have given experimental evidence of this apparently counterintuitive result?

Reviewer 2 Report

See attached.

Round 2

Reviewer 2 Report

The authors addressed my comments in a satisfactory manner; thus, I find the article suitable for publication in Polymers in its present form.